# Freeze Moisture Treatment and Ozonation of Adlay Starch (*Coix lacryma-jobi*): Effect on Functional, Pasting, and Physicochemical Properties

**DOI:** 10.3390/polym14183854

**Published:** 2022-09-15

**Authors:** Edy Subroto, Nisyrah Sitha, Fitry Filianty, Rossi Indiarto, Nandi Sukri

**Affiliations:** Department of Food Industrial Technology, Faculty of Agro-Industrial Technology, Universitas Padjadjaran, Bandung 45363, Indonesia

**Keywords:** adlay starch, freeze moisture treatment, ozonation, porous starch, starch modification

## Abstract

Adlay starch has great potential as a cereal starch, but it has several weaknesses, namely a low swelling volume, low solubility, and low stability. The purpose of this study was to improve the characteristics of adlay starch, such as porosity, functional properties, and pasting properties, through starch modification using freeze moisture treatment (FMT) and ozonation. This study consisted of several treatments, namely FMT, ozonation, and a combination of FMT + ozonation. The results show that the FMT and ozonation generally increased water absorption capacity, swelling volume, solubility, and number of pores of the starch granule. The pasting properties showed an increase in the viscosity of the hot paste and caused a decrease in the gelatinization temperature, breakdown, and setback viscosity. FMT 70% + ozonation produced modified adlay starch with a porous granular surface, swelling volume value of 21.10 mL/g, water absorption capacity of 1.54 g/g, a solubility of 9.20%, and an increase in the amorphous structure but did not cause the emergence of new functional groups. The combination of FMT + ozonation was effective in improving the functional, pasting, and physicochemical properties of adlay starch.

## 1. Introduction

Adlay is one of the cereals that can be used as a source of starch that can be applied to various food and chemical industries [1,2]. Starches from legumes and cereals generally have a high amylose content. Granules are small, oval in shape, and some are irregular [3,4,5]. However, native adlay starch has several weaknesses, including a low water absorption capacity, low swelling volume, and instable pasting properties [4,6]. Therefore, modification of native adlay starch is needed to improve the properties of starch so that it can be used more widely. Modification of starch and some treatments with the addition of other components, such as lipids and proteins, can also affect the physicochemical properties of starch [7,8,9,10].

Various methods for starch modification have been developed, namely chemical, physical, enzymatic, and microbiological modifications (fermentation) or a combination of several modification methods [11,12,13]. However, a modification process that is safe and does not cause hazardous chemical residues is preferred. Physical modifications using thermal processes that are safe have been developed, such as annealing (ANN), microwave heat treatment (MHT), and heat moisture treatment (HMT) [10,14,15,16]. However, the use of thermal modification by chilling or freezing to modify or improve the characteristics of starch is still limited. Modification by freezing may be possible through freeze moisture treatment (FMT). FMT is a new method that has not been widely used for starch modification. It utilizes freezing temperatures at a certain water content, which is then sublimated at low pressure (vacuum) using freeze drying. The sublimation process with a freeze dryer causes a reduction in water followed by the formation of a dry and porous product. The sublimation process due to the evaporation of ice crystals from the material will leave cavities or pores in which porous materials/starch may form [17,18]. Porous starch can be used as an absorbent, encapsulant, and for the manufacture of various instant products that require easy and fast absorption of water [19,20,21,22,23].

Modification of the FMT method can be combined with other methods to increase the effectiveness of improving the characteristics of adlay starch, such as oxidation with ozone (O_3_), known as ozonation. Ozone is a type of strong oxidizing agent but is safe for food products because it will not leave chemical residues as ozone will immediately turn back into oxygen after oxidizing the material [24,25,26]. Ozone can break the C-C bond on the glucose unit of starch so that it converts the hydroxyl groups into carbonyl groups and carboxyl groups [27,28]. Ozone and other oxidizing agents can change several functional properties of starch, such as increasing the whiteness degree, granule porosity, carboxyl content, solubility, water binding ability, and thermal stability [29,30].

FMT treatment combined with ozonation is expected to effectively improve the characteristics of adlay starch. Therefore, in this study, adlay starch was modified using FMT and ozonation methods, which were expected to improve starch characteristics such as functional properties, physicochemical properties, pasting properties, and starch granule morphology.

## 2. Materials and Methods

### 2.1. Materials

The “Mayuen” variety of adlay seed was obtained from Majalengka District, West Java, Indonesia. Other materials used were ozone, NaOH, distilled water, and some materials for analytical purposes, which were analytical-grade and purchased from Merck KGaA, Saint Louis, MO, USA.

### 2.2. Extraction of Adlay Starch

Adlay starch was extracted by a combination of dry and wet methods [4]. The dried adlay seeds were ground with a grinder until they became flour. The adlay flour was then soaked in NaOH solution (0.3%) for 24 h at a ratio of 1:3 (*w*/*v*), and then filtered. The dregs were immersed again in NaOH solution (0.3%) and then filtered up to two times. All the filtrate was deposited for one day, then washed with clean water, and then dried. The dried starch was then ground again and sifted at 100 mesh.

### 2.3. Modification of Adlay Starch by Freeze Moisture Treatment (FMT) and Ozonation

The experiment was carried on with five modified adlay starches and a control (native adlay starch). Modified adlay starches included ozonated adlay starch, FMT adlay starch at 60% initial moisture content, FMT adlay starch at the initial moisture content of 70%, FMT adlay starch at the initial moisture content of 60% + ozonation, and FMT starch adlay at the initial moisture content of 70% + ozonation. FMT was carried out in thin-wall containers, frozen at −50 °C, and sublimated using a freeze dryer (Christ Alpha 1–4 LDplus) at −50 °C for 48 h. Meanwhile, the ozonation was carried out using an ozone generator (D’Ozone DO150CB from PT. Dipo Technology, Indonesia, with an output capacity of 150 g/h) at a flow rate of 2 L/min for 20 min and stirred every 5 min [31].

### 2.4. Analysis of the Morphology of Starch Granule Particles

The surface morphology of starch was analyzed by SEM-JSM6510, JEOL Ltd. Tokyo, Japan. The sample was placed on the sample holder and then coated with gold. The morphology of the granules was read at magnification scales of 2000, 5000, and 7000.

### 2.5. Analysis of Functional Properties

Swelling volume (SV) and solubility were analyzed by weighing out 0.7 g of starch, which was then put into a centrifuge tube, to which 25 mL of distilled water was added, and finally, stirring the solution using a vortex. The sample was heated using a water bath (80 °C, 30 min). The sample was immersed in cold water and then centrifuged to separate the precipitate (gel) from the supernatant. The gel and supernatant in the centrifugation tube were separated, and then the supernatant volume was measured. The supernatant was then dried in a constant cup using an oven, then weighed [32], and SV was calculated using Formula (1), and solubility was calculated using Formula (2).

Water absorption capacity (WAC) was analyzed by weighing out one gram of adlay starch, to which 10 mL of distilled water was added, and then stirring the solution using a vortex, followed by incubated for 30–40 min and centrifugation to separate the precipitate from the supernatant. The supernatant was separated from the precipitate and weighed [33], and then WAC was calculated by Formula (3).
(1)$$\mathrm{Swelling}\;\mathrm{Volume}\;(\mathrm{SV})=\frac{\mathrm{Total}\;\mathrm{volume}\;(\mathrm{mL})\;-\;\mathrm{Supernatant}\;\mathrm{volume}\;(\mathrm{mL})}{\mathrm{Dry}\;\mathrm{sample}\;\mathrm{weight}\;(g)}$$
(2)$$\mathrm{Solubility}=\frac{\mathrm{Dry}\;\mathrm{supernatant}\;\left(g\right)}{\mathrm{Starch}\;\mathrm{sample}\;\left(g\right)}\;\times \;100\%$$
(3)$$\mathrm{WAC}=\frac{\mathrm{weight}\;\mathrm{of}\;\mathrm{distilled}\;\mathrm{water}\;\left(10\;g\right)-\mathrm{Weight}\;\mathrm{of}\;\mathrm{supernatant}\;\left(g\right)}{\mathrm{Weight}\;\mathrm{of}\;\mathrm{sampel}\;\left(1\;g\right)}$$

### 2.6. Analysis of Pasting Properties

A total of 3.5 g of starch was added to the distilled water (25 mL), and then a canister containing starch and aquadest was mounted on a Rapid Visco Analyzer (RVA-SM2). The test was carried out for 13 min, in which the temperature was raised to 95 °C and held for 7 min, and then the temperature was lowered to 50 °C and held for 5 min [14].

### 2.7. Analysis of Color

Color analysis was carried out using the Color Flex EZ CFEZ 2508 chromameter instrument. The chromameter was first calibrated with the standard black and white plates contained in the tool. The starch sample was put in a glass cup and then placed in the sample holder, and the color chromaticity of the sample was read. Tests were carried out with the Hunter L*, a*, b*, and ΔE* color system.

### 2.8. Analysis of Functional Groups

Functional groups of starch were analyzed with a “Nicolet iS10 FTIR spectrometer” with a spectral range of 7800 to 350 cm^−1^, a mid-infrared KBr beam splitter with a range of 11,000 to 375 cm^−1^, and XT KBr extended-range mid-infrared optics, combined with attenuated total reactance (ATR). The sample (1–2 mg) was added to pure KBr powder. The sample mixture was placed in the sample holder and then pressurized and maintained for several minutes. Then, the FTIR spectra were read, and the data were interpreted.

### 2.9. Statistical Analysis

The data obtained were analyzed for variance (ANOVA) with *p* < 0.05; then, to determine the differences between treatments, this step was followed by Duncan’s multiple range test using PASW Statistics version 18.0 (IBM Company, Armonk, NY, USA).

## 3. Results and Discussion

### 3.1. Swelling Volume (SV) and Solubility

SV and solubility are affected by the noncovalent bonds of amylose and amylopectin. Starch granules will absorb water and swell to about 30% of the initial volume when starch is put into cold water. However, when starch granules are heated in excess water, it causes the breaking of hydrogen bonds and disruption of the crystalline structure so that starch granules expand. The SV of native adlay starch and adlay starch modified by FMT and ozonation is shown in Figure 1, and the solubility can be seen in Figure 2.

Figure 1 shows that the treatments by FMT 60%, FMT 70%, FMT 60% + ozonation, and FMT 70% + ozonation increased the SV of adlay starch compared to that of native starch (*p* < 0.05). Meanwhile, the ozonation treatment without FMT did not increase SV significantly. The highest increase occurred in the starch modified by FMT 60% and FMT 70%, whose SV increased from 15.98 mL/g to 22.94 mL/g or up to 1.44 times higher than that of native starch. The increase in the FMT-modified starch could have been caused by the formation of cavities in the starch granules due to the sublimation of ice crystals in the starch granules, which allowed the starch to absorb and bind more water, leading to its expansion. Meanwhile, the increase in the SV value of FMT + ozonated-treated starch could have been caused by the formation of cavities and the increase in carbonyl (-CO) and carboxyl (-COOH) groups during the oxidation process [34]; the schematic reaction of starch oxidation can be seen in Figure 3. Depolymerization of amylose chains into carbonyl and carboxyl groups caused water components to enter the amylopectin chain and increase the swelling power of starch. This can be explained by the fact that amylose is more easily broken down in the early stages of oxidation, which has implications for depolymerization of the amylose chain, and the water molecules contained in the system are easily accessed by amylopectin molecules, causing an increase in the swelling volume [35,36].

Based on Figure 2, the treatments by ozonation, FMT 60%, FMT 70%, FMT 60% + ozonation, and FMT 70% + ozonation increased the solubility of adlay starch significantly (*p* < 0.05) compared to that of native starch. The largest increase occurred in adlay starch modified by FMT 60%, FMT 70%, FMT 60% + ozonation, and FMT 70% + ozonation, which exhibited an increase up to 1.44 times higher that of native starch. The increase in the solubility of FMT-modified starch and ozonation was the result of depolymerization of starch molecules, which increased along with the weakness of the granule structure due to ice crystallization in the granules, followed by sublimation. The depolymerization was also caused by the formation of carbonyl (-CO) and carboxyl (-COOH) groups during the oxidation, as shown in the schematic diagram of the oxidation of the starch reaction (Figure 3). The oxidation of C-C carbon bonds caused the depolymerization of amylose and increased the solubility of the amylose fraction in water [37].

### 3.2. Water Absorption Capacity (WAC)

WAC is the ability of starch to hold on to water that is absorbed. The WAC value appertains to the granule composition and physical properties of adlay starch after the addition of water. The WAC of native adlay starch and adlay starch modified by FMT and ozonation can be seen in Figure 3.

Based on Figure 4, the WAC in adlay starch modified by FMT and ozonation showed an increase. The treatments by ozonation, FMT 60%, FMT 70%, FMT 60% + ozonation, and FMT 70% + ozonation increased the solubility of adlay starch significantly (*p* < 0.05) compared to that of native starch, with the highest increase occurring in adlay starch modified by FMT 60% + ozonation and FMT 70% + ozonation, which exhibited an increase in solubility up to 1.26 times higher than that of native starch. The increase in the WAC in adlay starch modified by FMT and ozonation was caused by the formation of pores or cavities due to water crystallization in the granules followed by sublimation, leading to a higher starch surface area and the absorption of more water. The more porous the material, the higher the water absorption rate. The WAC in starch was also affected by the presence of hydroxyl and carboxyl groups in starch molecules. The oxidation of starch by ozonation splits the C-C bond into a carboxyl group. When the number of hydroxyl and carboxyl groups in starch molecules is large, the ability to absorb water increases [38,39]. The ozonation reaction is also thought to cause a partial breakdown of the branching chain of the amylopectin molecule, which causes it to turn into a straight-chain molecule so that the amount of short-chain amylose increases. The short-chain amylose molecule is hydrophilic and can dissolve in water.

### 3.3. Color

Color chromaticity is expressed in values of L*, a*, and b*, while the total color change is expressed as ΔE*. The value of L* indicates the brightness level, which has a scale of 1–100 from black to white; the value of a* indicates the presence of red if positive (+) and the presence of green if negative (−), while the value of b* indicates the color yellow if positive (+) and indicates blue if negative (−). The color of native adlay starch and adlay starch modified by FMT and ozonation can be seen in Table 1.

Table 1 shows that the color of all adlay starch treatments showed only a slight change. The brightness level (L*) in all treatments had a positive value ranging from 93.50 to 94.20, which indicated that all treatments of adlay starch were bright white. However, the treatments of FMT 60% and FMT 70% decreased the L* (*p* < 0.05). The values of a* were in the range of 0.15–0.20, where the treatments of FMT 60%, FMT 70%, and FMT 70% + ozonation decreased the a* (*p* < 0.05). The values of b* were in the range of 3.65–4.21, where the treatments of FMT 60% increased the b* (*p* < 0.05). While the values of ΔE* of modified adlay starch were in the range of 0.15–0.63, indicating that the total color change was slight, all treatments increased the ΔE* (*p* < 0.05). Modification of FMT used a low temperature of −50 °C so as to prevent damage or coloration caused by reactions triggered by high temperatures. Meanwhile, the oxidation reaction induced by the ozonation process that occurs can actually cause some of the color-forming pigments to be oxidized before glucose units so that some of the existing compounds will be lost, and modified adlay starch was whiter, brighter, and exhibited a more intensive reduction in the intensity of the yellow color in starch [40].

### 3.4. Pasting Properties

The pasting properties are expressed in the form of a visco-amylograph that can be seen in Figure 5, while the data for the observed pasting properties’ parameters are shown in Table 2. Figure 5 and Table 2 show that the modification of FMT and ozonation caused the pasting point of adlay starch to decrease, where the largest decrease occurred in adlay starch modified by FMT 70% + ozonation, which showed a decrease from 76.92 °C to 76.02 °C. The decrease in the pasting point was due to the FMT treatment increasing the pores and cavities in the starch granule particles, making it easier for the starch granules to absorb water when heated and gelatinize more quickly. Meanwhile, ozonation caused oxidation, which resulted in a weakening of the starch structure, then increased the starch sensitivity to temperature, making the gelatinization of the starch granules easier and faster. In general, the peak viscosity decreased in FMT-and-ozonation-modified starch compared to native starch, where the largest decrease occurred in ozonated adlay starch, which showed a decrease from 5687.50 cP to 5214.67 cP. This was caused by the partial breakdown of the glycosidic chain during oxidation, preventing the starch from maintaining its integrity. The carboxyl group in the starch structure weakened the starch granule structure and contributed to lowering the peak viscosity [41,42].

Parameters of hot paste viscosity (hold viscosity) and breakdown viscosity were correlated to each other because breakdown viscosity was the deviation between peak viscosity and hold viscosity. Based on Figure 5 and Table 2, it was found that the starch modified by FMT 60% + ozonation, and FMT 70% + ozonation had a decreased pasting point (*p* < 0.05) compared to native adlay starch. This indicated that the granules of modified starch were more easily gelatinized due to the formation of cavities after the sublimation of ice crystals during FMT and depolymerization due to oxidiziation by ozonation treatment. Ozonation treatment also decreased the peak viscosity (*p* < 0.05), while other treatments had no effect on peak viscosity. However, FMT 70%, FMT 60% + ozonation, and FMT 70% + ozonation could increase the hold viscosity (*p* < 0.05) of the modified starch compared to native adlay starch. The highest increase in hold viscosity occurred in FMT 70% + ozonation, where the hold viscosity increased from 1322 cP to 1502.33 cP. The changes in viscosity during heating or breakdown indicated the stability of the paste during heating, where the lower the breakdown viscosity, the more stable the paste formed against heat. Figure 5 and Table 2 show that the modification of ozonation and FMT 70% + ozonation reduced the breakdown viscosity of adlay starch (*p* < 0.05). This indicated that the starch modified by ozonation and FMT 70% + ozonation had better heat stability than the native adlay starch. This could be due to the increased regularity of the crystalline matrix due to the large amount of amylose formed during the FMT and oxidation by ozonation, and the formation of amylose complexes with other components could improve the stability of the paste during heating [43].

Modification of adlay starch by FMT and ozonation (all treatments) did not affect the cold paste viscosity (final viscosity). However, the treatments of ozonation and FMT 70% + ozonation caused a decrease in setback viscosity (*p* < 0.05). This indicated that starch modified by the treatments of ozonation and FMT 70% + ozonation had higher stability when refrigerated and had a lower tendency towards retrogradation than native adlay starch. The decrease in setback viscosity was due to FMT and oxidation facilitating the depolymerization of amylose and amylopectin, which decreased the viscosity and retrogradation rate. The structural rearrangement of carbonyl and carboxyl in oxidized starch made the paste more susceptible to association tendencies during cooling and decreased the setback viscosity [36,44].

### 3.5. Starch Granule Morphology

The particle morphology of native adlay starch and adlay starch modified by FMT and ozonation can be seen in Figure 6. It shows that the surface structure of starch tends to be polygonal and irregularly round. The results of the SEM analysis on natural adlay starch show its smooth and intact surface, while for the starch modified by ozonation, FMT, and the combination of FMT + ozonation, the surface becomes rougher and more porous than that of native adlay starch. If we look closely, the best treatment was the FMT 70% + ozonation treatment, which produced starch granules that had many pores, were hollow, and had a rough surface. The results of this study are in line with the research of Çatal and Ibanoǧlu [45] on ozonated corn and potato starch, which showed that the surface became rough and porous after the ozonation treatment. It was suspected that oxidation caused a weakening of the surface structure of the starch. Most of the oxidation occurred on the outer granule surface because the surface of the starch molecules was more easily oxidized than that of the crystals due to their higher accessibility [38,46].

Figure 6 also shows that the FMT modification also increased the pores and cavities in the granules. This was caused by the release of water vapor during the sublimation of ice crystals in the starch granules, resulting in distortion of the starch granules when the vacuum conditions pushed water molecules out through the dense internal structure of the material granules. This condition increased the number of pores on the surface of the starch granules, which led to their high water absorption capacity. This was in line with Xie et al. [47], who showed that the surface of wheat, potato, and pea starch grains that underwent sublimation treatment had more pores and scratches than starch treated with oven drying.

### 3.6. Functional Groups of Starch

The functional groups of starch as seen in the FTIR-ATR spectra are presented in Figure 7, and the FTIR spectra band assignment can be seen in Table 3. Based on Figure 7 and Table 3, the FTIR-ATR spectra of native adlay starch and starch modified by FMT 70% + ozonation showed the presence of O-H groups with a wave peak of around 3397–3398 cm^−1^. The next wave peak was about 2929–2930 cm^−1^, indicating the presence of a C-H bond vibration. The next peak indicates the presence of a C=O double bond with a wave number of around 1640–1643 cm^−1^, which indicates the presence of carbonyl groups. The next peak indicates the presence of a C–O bond with a wave number of around 1304–1305 cm^−1^, which indicates the presence of carboxyl groups. The FTIR spectrum of native adlay starch and starch modified by FMT 70% + ozonation had the same pattern, and no new spectral peaks were found. This indicated that there was no change in or formation of new functional groups, but there was an increase in the intensity of the hydroxyl and carbonyl groups in the modified starch. The increase in the hydroxyl and carbonyl functional groups in the modified starch indicated a greater ability to absorb water, as well as confirmed the swelling volume in Figure 1 and water absorption capacity in Figure 4. The binding ability and water absorption capacity increase with more hydroxyl groups in the starch molecule [38,39].

FTIR-ATR spectra also show the microstructure of starch, especially the crystalline and amorphous structures. The intensity ratio for wavenumbers 1045/1022 indicates a crystalline structure, while the intensity ratio for wavenumbers 1022/995 indicates an amorphous structure [48]. Based on the FTIR-ATR spectra and the calculation results, the absorbance ratio value of 1045/1022 in adlay starch modified by FMT 70% + ozonation only slightly increased from 0.90 to 0.91. Meanwhile, the value of the absorbance ratio of 1022/995 showed that the adlay starch modified by FMT 70% + ozonation increased to 1.24 compared to that of native adlay starch, with a value of 1.19. This indicated that the FMT 70% + ozonation-modified starch had a larger amorphous structure than native starch. This was also confirmed in Figure 6, which shows that the particles of modified adlay starch had more pores and cavities compared to native starch.

Based on the results and discussion, the significance of this study can be explained as follows: The treatments by FMT and ozonation could significantly improve the functional properties of adlay starch, as seen in the swelling volume, solubility, and water absorption capacity. The treatments also affected the pasting properties by increasing the stability of the starch paste during heating and cooling. The treatments can also produce porous starch granules that can potentially be used for various food, chemical, and pharmaceutical products.

## 4. Conclusions

Modification of FMT and ozonation of adlay starch was able to increase water absorption capacity, swelling volume and solubility, and increase the number of pores and amorphous structures without the formation of new functional groups. The modified starch also showed an increase in the hot paste viscosity and a decrease in the gelatinization temperature, peak viscosity, hold viscosity, breakdown viscosity, and setback viscosity. Treatment by FMT 70% + ozonation gave a significant effect by producing modified adlay starch with a swelling volume of 21.10 mL/g, a water absorption capacity of 1.54 g/g, and a solubility of 9.20%, which was also more stable under heating conditions, had porous starch granules, and exhibited an increase in the amorphous structure, proving its ability to improve the functional and physicochemical properties of adlay starch.

## Figures and Tables

**Figure 1 polymers-14-03854-f001:**
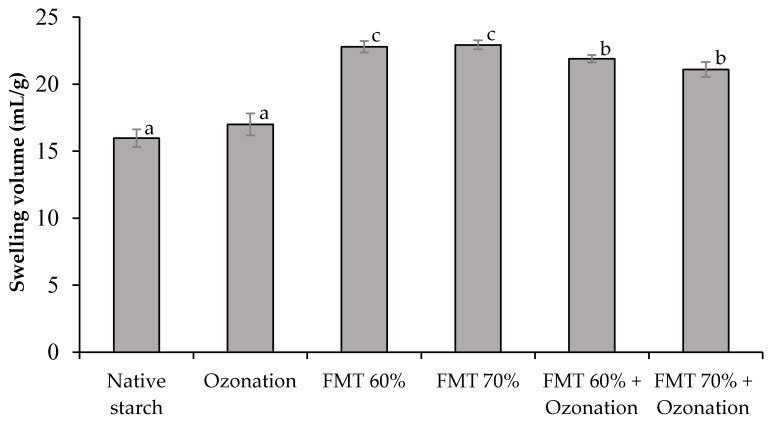
The swelling volume of native adlay starch and adlay starch modified by FMT and ozonation. Different letters show significant differences between treatments (*p* < 0.05).

**Figure 2 polymers-14-03854-f002:**
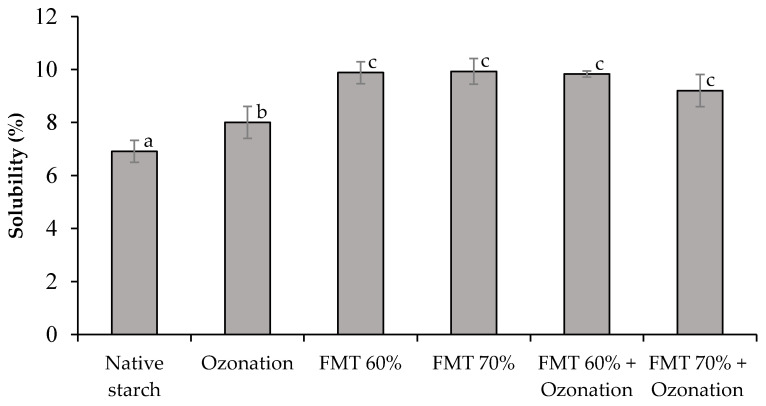
The solubility of native adlay starch and adlay starch modified by FMT and ozonation. Different letters show significant differences between treatments (*p* < 0.05).

**Figure 3 polymers-14-03854-f003:**
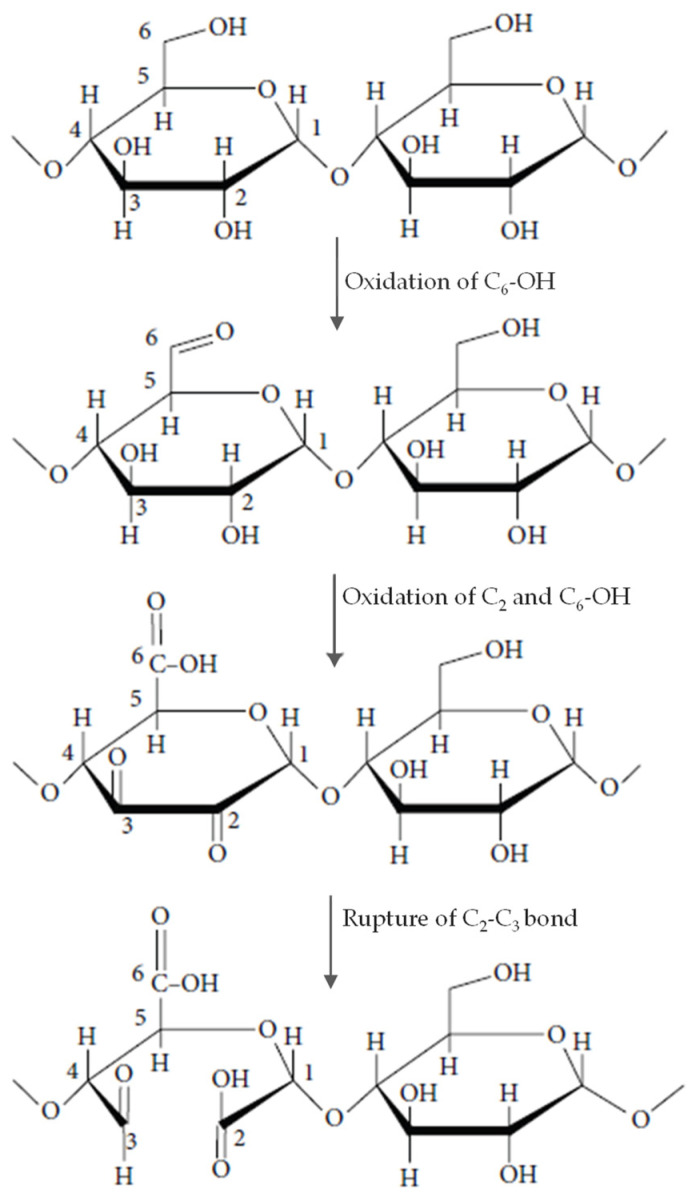
Schematic diagram of carbonyl and hydroxyl groups formed in starch oxidation.

**Figure 4 polymers-14-03854-f004:**
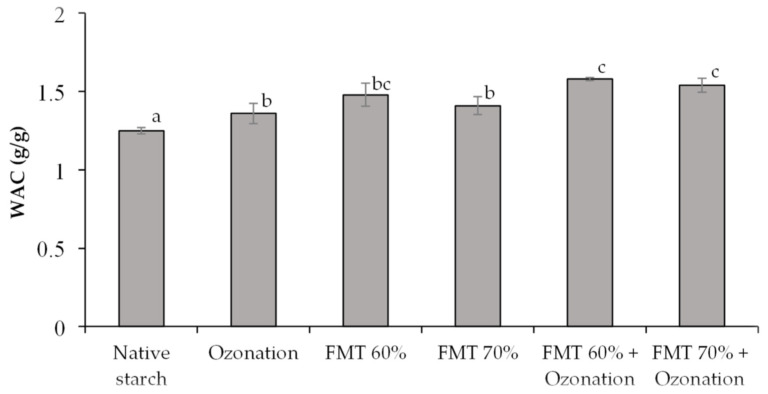
The water absorption capacity (WAC) of native adlay starch and adlay starch modified by FMT and ozonation. Different letters show significant differences between treatments (*p* < 0.05).

**Figure 5 polymers-14-03854-f005:**
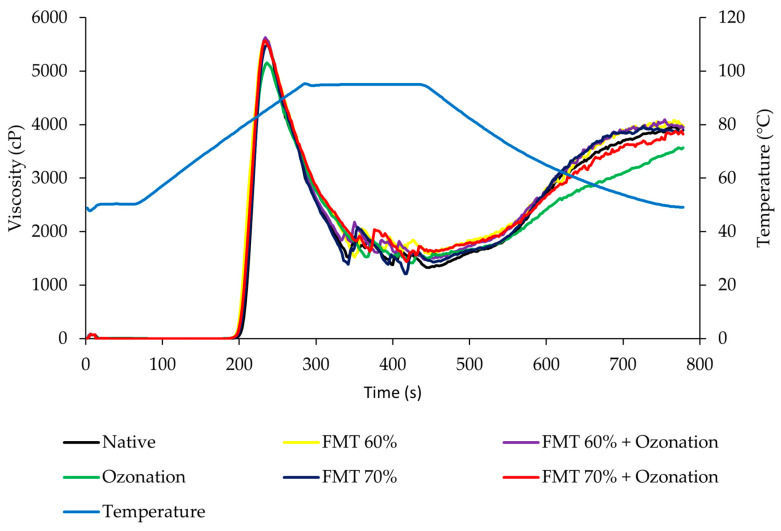
The visco-amylograph of native adlay starch and adlay starch modified by FMT and ozonation.

**Figure 6 polymers-14-03854-f006:**
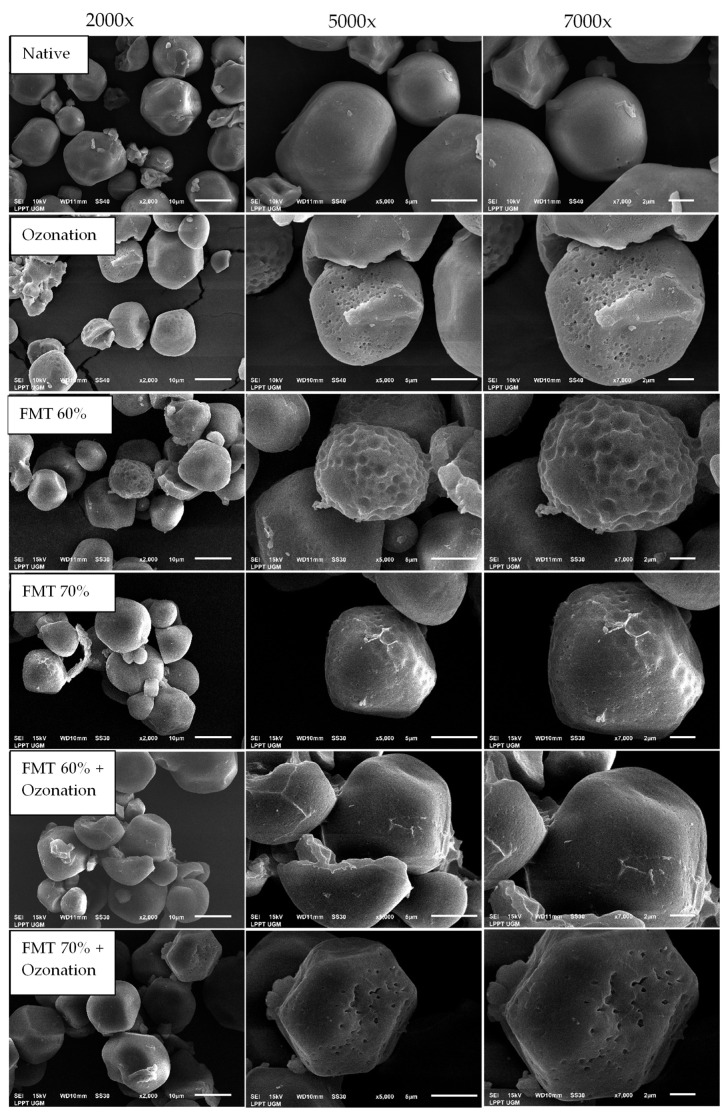
The particle morphology of native adlay starch and adlay starch modified by FMT and ozonation.

**Figure 7 polymers-14-03854-f007:**
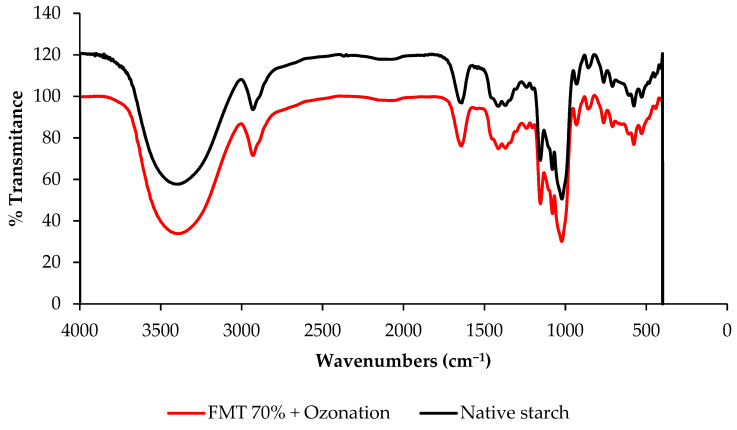
The FTIR-ATR spectra of native adlay starch and adlay starch modified by FMT 70% + ozonation.

**Table 1 polymers-14-03854-t001:** The color of native adlay starch and adlay starch modified by FMT and ozonation.

Treatment	L*	a*	b*	ΔE*
Native starch	94.05 ± 0.30 ^a^	0.19 ± 0.02 ^c^	3.88 ± 0.08 ^a^	0.00 ± 0.00 ^a^
Ozonation	93.56 ± 0.65 ^ab^	0.20 ± 0.01 ^c^	3.65 ± 0.17 ^a^	0.54 ± 0.03 ^c^
FMT 60%	93.51 ± 0.03 ^b^	0.17 ± 0.01 ^b^	4.21 ± 0.08 ^b^	0.63 ± 0.05 ^c^
FMT 70%	93.50 ± 0.12 ^b^	0.15 ± 0.01 ^a^	3.95 ± 0.05 ^a^	0.56 ± 0.05 ^c^
FMT 60% + ozonation	94.20 ± 0.03 ^a^	0.20 ± 0.01 ^c^	3.86 ± 0.06 ^a^	0.15 ± 0.01 ^b^
FMT 70% + ozonation	94.02 ± 0.13 ^a^	0.17 ± 0.03 ^b^	3.71 ± 0.17 ^a^	0.17 ± 0.01 ^b^

Different letters in the same column indicate a significant difference (*p* < 0.05).

**Table 2 polymers-14-03854-t002:** The pasting properties of native adlay starch and adlay starch modified by FMT and ozonation.

Treatment	Pasting Point (°C)	Peak Viscosity (cP)	Hold Viscosity (cP)	Final Viscosity (cP)	Breakdown Viscosity (cP)	Setback Viscosity (cP)
Native	76.92 ± 0.25 ^ab^	5687.50 ± 163.34 ^a^	1322.00 ± 7.07 ^a^	3918.50 ± 70.00 ^a^	4365.50 ± 156.27 ^a^	2596.50 ± 62.93 ^a^
Ozonation	76.47 ± 0.63 ^abc^	5214.67 ± 187.09 ^b^	1394.00 ± 109.30 ^ab^	3565.00 ± 332.45 ^a^	3820.67 ± 268.19 ^b^	2171.00 ± 260.48 ^b^
FMT 60%	76.38 ± 0.28 ^bc^	5587.00 ± 102.27 ^a^	1194.33 ± 227.98 ^ab^	3885.00 ± 171.50 ^a^	4392.67 ± 222.78 ^a^	2690.67 ± 322.06 ^ab^
FMT 70%	76.58 ± 0.48 ^abc^	5512.67 ± 71.44 ^a^	1401.00 ± 56.63 ^b^	3892.33 ± 71.65 ^a^	4111.67 ± 127.01 ^ab^	2491.33 ± 126.18 ^ab^
FMT 60% + Ozonation	75.99 ± 0.39 ^c^	5675.33 ± 40.81 ^a^	1453.00 ± 35.79 ^b^	3943.00 ± 46.29 ^a^	4222.33 ± 36.23 ^a^	2490.00 ± 28.58 ^a^
FMT 70% + Ozonation	76.02 ± 0.29 ^c^	5561.33 ± 116.52 ^a^	1502.33 ± 111.59 ^b^	3819.67 ± 147.82 ^a^	4079.00 ± 46.18 ^b^	2317.33 ± 129.11 ^b^

Different letters in the same column indicate a significant difference (*p* < 0.05).

**Table 3 polymers-14-03854-t003:** FTIR-ATR spectra band assignment for adlay starch.

Band Assignment	Wavenumbers (cm^−1^)
O–H stretch	3397–3398
C–H stretch	2929–2930
C–C stretch	1412–1413
C=O stretch (Carbonyl)	1640–1643
C–O stretch (Carboxyl)	1304–1305

## Data Availability

Not applicable.

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
