# Peer review of "Freeze Moisture Treatment and Ozonation of Adlay Starch (Coix lacryma-jobi): Effect on Functional, Pasting, and Physicochemical Properties"

_polymers, 2022, doi:10.3390/polym14183854_

Round 1

Reviewer 1 Report

This study investigated the effects of freeze moisture treatment and ozonation on the functional, pasting, and physicochemical properties of the Adlay Starch (Coix lacryma-jobi). The object of this work is interesting; However, the results were not statistically analyzed. Therefore, it is not acceptable in its present form.

Comments:

1. L 14, 15, and 21; It is better to use “FMT” instead of “freeze moisture treatment”.

2. L 15; “The results showed that the treatment increased …”, Which treatment?

2. You should elaborate more on the basic information and importance of the starches in the introduction section;

DOI: 10.3390/polym14153012

DOI: 10.1111/ijfs.16021

3. L 33; What did the authors mean by the microbiological modifications of the starch?

4. L 50; O3, not O3

5. L 63, (Section 2.1); The authors should elaborate more on the material’s details.

6. L 74; You should not consider the control treatment (native starch) as a modified treatment. So, the experiment was carried on with five modified starches and a control.

7. L 80; Please mention more details about an ozone generator.

8. L 111; Please add more details about the FTIR experiment, such as resolution, spectra range, etc.

9. At the end of the Materials and Methods Section, the authors should add the “Statistical Analysis” part. After that, all tables and figures should be statistically analyzed, and the significant letters should be mentioned. Then, all results should be re-evaluated and rewritten (Sections 3.1, 3.2, 3.3, and 3.4).

10. L 263, Section 3.5; cm-1, not cm-1

11. L 277; Figure 6, not Figure 4

Author Response

Response to Reviewer 1 Comments

This study investigated the effects of freeze moisture treatment and ozonation on the functional, pasting, and physicochemical properties of the Adlay Starch (Coix lacryma-jobi). The object of this work is interesting; However, the results were not statistically analyzed. Therefore, it is not acceptable in its present form.

Point 1: L 14, 15, and 21; It is better to use “FMT” instead of “freeze moisture treatment”.

Response 1:

“freeze moisture treatment” has been revised to “FMT” (Page 1, Line 14, 15, and 21, red color).

Point 2: L 15; “The results showed that the treatment increased …”, Which treatment?

Response 2:

"Which treatment" means FMT and ozonation. “the treatment” has been revised to “FMT and ozonation” (Page 1, Line 15-16, red color).

Point 3: You should elaborate more on the basic information and importance of the starches in the introduction section;

DOI: 10.3390/polym14153012

DOI: 10.1111/ijfs.16021

Response 3:

The basic information and importance of the starches have been elaborated in the introduction section. References from DOI: 10.3390/polym14153012 and DOI: 10.1111/ijfs.16021 have been cited (Page 1, Line 27-34, and Page 13, Line 381-387, red color).

Point 4: L 33; What did the authors mean by the microbiological modifications of the starch?

Response 4:

microbiological modifications of the starch mean fermentation. This information has been added to the sentence (Page 1, Line 36, red color).

Point 5: L 50; O3, not O3

Response 5:

O3 has been revised to O3 (Page 2, Line 53, red color)

Point 6: L 63, (Section 2.1); The authors should elaborate more on the material’s details.

Response 6:

The material’s details have been elaborated (Page 2, Line 67-70, red color).

Point 7: L 74; You should not consider the control treatment (native starch) as a modified treatment. So, the experiment was carried on with five modified starches and a control.

Response 7:

The experiment has been revised that the experiment was carried on with five modified starches and a control (Page 2, Line 79-83, red color).

Point 8: L 80; Please mention more details about an ozone generator.

Response 8:

The details about an ozone generator have been mentioned (Page 2, Line 85-87, red color).

Point 9: L 111; Please add more details about the FTIR experiment, such as resolution, spectra range, etc.

Response 9:

The details about the FTIR experiment, such as resolution, spectra range, etc., have been added (Page 3, Line 118-120, red color).

Point 10: At the end of the Materials and Methods Section, the authors should add the “Statistical Analysis” part. After that, all tables and figures should be statistically analyzed, and the significant letters should be mentioned. Then, all results should be re-evaluated and rewritten (Sections 3.1, 3.2, 3.3, and 3.4).

Response 10:

The “Statistical Analysis” has been added at the end of the Materials and Methods Section (Page 3, Line 124-127, red color). All tables and figures have been statistically analyzed, and the significant letters have been mentioned (Table 1 and Table 2, Figure 1, Figure 2, Figure 4, red color) . All results also have been re-evaluated and rewritten (Page 4-9, red color).

Point 11: L 263, Section 3.5; cm-1, not cm-1

Response 11:

Section 3.5; cm-1 has been revised to cm-1 (Page 11, Line 311, 312, and 314, red color).

Point 12: L 277; Figure 6, not Figure 4

Response 12:

L 277; Figure 4 has been revised to Figure 7 (there is an addition of Figure 3) (Page 6, Line 308-309, and 325, red color).

Reviewer 2 Report

This manuscript is a solid scientific research paper. It is sufficient and concise. The writing style is good, and the organization is logical. I only have several recommendations that hopefully can help improve the manuscript. 

1. Add schematics of the chemical reactions for better clarification and visualization.  

2. It is better to add a table of FTIR peaks and corresponding functional groups in section 3.5. 

3. In line 283, "from 0.90 to 091" I think this is a typo. Please correct. 

4. Add one paragraph or section of the significance of this study. 

5. The authors cited half of the reference in Introduction, another half in the Results and Discussion. More reference should be cited in Introduction and the results and discussion should focus more on data and discussion around data. 

Author Response

Response to Reviewer 2 Comments

This manuscript is a solid scientific research paper. It is sufficient and concise. The writing style is good, and the organization is logical. I only have several recommendations that hopefully can help improve the manuscript.

Point 1: 1. Add schematics of the chemical reactions for better clarification and visualization.

Response 1:

The schematics of the chemical reactions of starch oxidation have been added for better clarification and visualization (Page 5, Figure 3, Line 173-177).

Point 2: It is better to add a table of FTIR peaks and corresponding functional groups in section 3.5.

Response 2:

A table of FTIR peaks and corresponding functional groups have been added in section 3.5 (Page 11, Table 3, Line 327-328, red color).

Point 3: In line 283, "from 0.90 to 091" I think this is a typo. Please correct.

Response 3:

Line 283, "from 0.90 to 091" has been corrected to “from 0.90 to 0.91" (Page 12, Line 334, red color).

Point 4: Add one paragraph or section of the significance of this study.

Response 4:

A paragraph or section of the significance of this study has been added (Page 12, Line 340-346, red color).

Point 5: The authors cited half of the reference in Introduction, another half in the Results and Discussion. More reference should be cited in Introduction and the results and discussion should focus more on data and discussion around data.

Response 5:

More references have been cited in the Introduction, and the results and discussion have been revised to focus on data and discussion around data.

Round 2

Reviewer 1 Report

The authors addressed all my suggestions. This manuscript can be accepted after a minor revision:

1. Please use the letter "a" for the largest value.

2. For Figures 1, 2, and 4; Please place "Different letters showed significant differences between treatments (p < 0.05)" after the figures' caption.